# Revisiting regression methods for estimating long-term trends in sea surface temperature

Ming-Huei Chang[1,2,*], Yen-Chen Huang[1], Yu-Hsin Cheng[3], Chuen-Teyr Terng[4], Jinyi Chen[4], Jyh Cherng Jan[4]

[1]Institute of Oceanography, National Taiwan University, Taipei, 10617, Taiwan
[2]Ocean Center, National Taiwan University, Taipei, 10617, Taiwan
[3]Department of Marine Environmental Informatics, National Taiwan Ocean University, Keelung, 202301, Taiwan
[4]Central Weather Administration, Taipei, 10617, Taiwan

*Correspondence to: Ming-Huei Chang (minghueichang@ntu.edu.tw)

**Abstract.** Global warming has enduring consequences in the ocean, leading to increased sea surface temperatures (SSTs) and subsequent environmental impacts, including coral bleaching and intensified tropical storms. It is imperative to monitor these trends to enable informed decision-making and adaptation. In this study, we comprehensively examine the methods for extracting long-term temperature trends, including the seasonal-trend decomposition procedure based on loess (STL) and the linear regression family, which comprises ordinary least square regression (OLSR), orthogonal regression (OR), and geometric

mean regression (GMR). The applicability and limitations of these methods are assessed based on experimental and simulated data. STL may stand out as the most accurate method for extracting long-term trends. However, it is associated with a notably sizeable computational time. In contrast, linear regression methods are far more efficient. Among these methods, GMR is not suitable due to its inherent assumption of a random temporal component. OLSR and OR are preferable for general tasks but require correction to accurately account for seasonal signal-induced bias resulting from the phase-distance imbalance. We

observe that this bias can be effectively addressed by trimming the SST data to ensure that the time series becomes an even function before applying linear regression, which is termed evenization. We compare our methods with two commonly used methods in the climate community. Our proposed method is unbiased and better than the conventional SST anomalies method. While our method may have a larger degree of uncertainty than the combined linear and sinusoidal fitting, this uncertainty remains within an acceptable range. Furthermore, linear and sinusoidal fitting can be unstable when applied to natural data

containing significant noise.

## 1 Introduction

Global warming refers to the long-term increase in the Earth's average surface temperature due to the accumulation of greenhouse gases in the atmosphere, including carbon dioxide, methane, and nitrous oxide. Global warming has long-term consequences in the ocean, particularly rising sea levels and increased sea surface temperatures (SSTs). The former is mainly

driven by the expansion of seawater due to higher temperatures and the melting of land ice. Elevated sea levels pose risks to

coastal communities, infrastructure, and ecosystems (Hinkel et al., 2010; Le Cozannet et al., 2014; Ranasinghe, 2016) due to the increased threat of coastal flooding, erosion, and saltwater intrusion. Moreover, global warming has led to increased sea surface temperatures. Rising sea temperatures have the potential to cause changes in ocean circulation patterns. Research has shown that the Kuroshio and Gulf Stream, two important subtropical western boundary currents in the North Pacific and North

Atlantic, can become stronger (Sakamoto et al., 2005; Cheon et al., 2012; Chen et al., 2019; Wang and Wu, 2019) and weaker (Levermann et al., 2005; Chen et al., 2019), respectively. This can ultimately impact the Atlantic meridional overturning circulation (AMOC), as the Gulf Stream is a key system component. The impact of SST warming on tropical cyclones has been a top concern in recent decades (Emanuel, 2005). As global warming continues, we see fewer cyclones overall, but those that do occur are more powerful, longer-lasting, larger, and more destructive (Emanuel, 2005; Maue et al., 2011; Lin et al.,

2014; Sun et al., 2017). This increase in destructive potential is due to the combination of longer storm lifetimes and greater storm intensities resulting from warmer sea surface temperatures. However, the situation may be more nuanced, as other atmospheric conditions, such as increased wind shear, could counteract or even reverse this trend of heightened destruction (Lin and Chan, 2015). Coral reefs are facing an increasing threat due to rising ocean temperatures (Pandolfi et al., 2011). This has resulted in the unprecedented mass bleaching of corals, which has been triggered by rising sea surface temperatures (Frieler

et al., 2013; Hughes et al., 2017; Hoegh-Guldberg et al., 2017; Sully et al., 2019). Although some mitigations have been observed through small-scale local upwelled or mixed cold water (Tkachenko and Soong, 2017; Safaie et al., 2018; Davis et al., 2021), the overall situation remains concerning.

Therefore, studying and monitoring these long-term change trends is essential to understand and address the challenges associated with global warming. It is crucial for decision-making and adaptation strategies (Mimura, 2013). Trends in sea

surface temperature changes are obtained by analyzing long-term data collected from various sources, including satellite remote sensing, buoys, ships, and coastal monitoring stations. This approach involves methods such as linear regression to determine the temperature change slope over a specific period. Such data could generally be decomposed as (Cleveland et al., 1990)

$$ST = ST_{LT} + ST_{SV} + ST_R. \tag{1}$$

where $ST$ is the measured SST, and $ST_{LT}$, $ST_{SV}$, and $ST_R$ are the long-term trend, seasonal variations, and remainder component, respectively. $ST_{SV}$ is often the most predominant component in the collected data $ST$. $ST_R$ primarily encompasses the signals of tides and minor signals of subseasonal variations (e.g., mesoscale and submesoscale processes and inertial oscillations), day-night variation (e.g. Chang et al., 2023), and unresolved noises (measuring error and turbulence). Commonly used methods to extract $ST_{LT}$ include linear regression methods (Emery and Thomson, 2001; Boretti, 2020; Sreeraj et al., 2022) and the

seasonal-trend decomposition procedure based on loess (STL; Cleveland et al., 1990; Tseng et al., 2010; Nidheesh et al., 2013), where loess refers to locally estimated scatterplot smoothing (Cleveland and Devlin, 1988). Linear regression is a widely used and efficient statistical technique to model the relationship between a dependent variable (in this case, sea surface temperature) and an independent variable (typically time). STL is a robust and versatile method for decomposing time series data into long-term trends, seasonal variations, and residuals. STL algorithms use Loess to smooth the data and extract the long-term trend

65 component, which has been recognized as a better method to extract the trend (Tseng et al., 2010; Nidheesh et al., 2013). However, long computational time is a primary concern when employing STL. Moreover, while the STL method typically captures a nonlinear trend, many practical applications necessitate a linear trend to depict the overall scenario better. Linear regression methods have been a common choice for extracting the long-term trend of SST increase. These methods are often applied without thorough assessment due to their universal nature, which is well documented in statistics textbooks. However,

70 the long-recorded SST data possess unique characteristics that could introduce bias, requiring careful attention to the details of the analysis. Commencing with the fundamentals of linear regression and utilizing realistic and simulated data, this study offers a systematic evaluation and comparison of linear regression methods and STL.

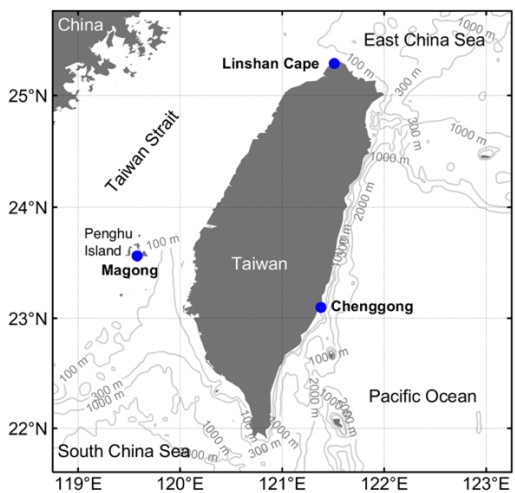

**Figure 1: Topography and coastal line surrounding Taiwan. The blue dots denote three coastal buoys at Chenggong, Linshan Cape,**
75 **and Magong stations, maintained by Taiwan's Central Weather Administration (CWA).**

## 2 Methodology

Linear regression analysis begins with considering the dependent variable $y$ and the independent variable $x$. $y$ and $x$ can be decomposed as $y = Y + \varepsilon$ and $x = X + \delta$, respectively. Here, $(X, Y)$ and $(\delta, \varepsilon)$ represent the deterministic and random components, respectively, and the mean values of $\varepsilon$ and $\delta$ are 0 ($\mu_\varepsilon = \mu_\delta = 0$). Linear regression seeks to determine a linear

80 relationship between their deterministic part via the model $Y = b_0 + b_1 X$ by determining the value of $b_0$ $and$ $b_1$. The presence of the random component is undesirable and significantly affects the extraction of the linear relationship within their deterministic part. Here, the deterministic component of $y$ is $ST_{LT}$, and the random component is intended to mimic $ST_{SV} + ST_R$. Note that $ST_{SV}$ is indeed non-random, which could bias the estimate of $b_1$ and $b_0$. Based on the likelihood estimate (Wong, 1989; Emery and Thomson, 2001; Leng et al., 2007), the slope of the model $b_1$ can be estimated as

85 $$\widehat{b_1} = \frac{S_{yy} - \lambda S_{xx} + \sqrt{(S_{yy} - \lambda S_{xx})^2 + 4\lambda S_{xy}^2}}{2 S_{xy}},$$  (2)

where $S_{xx}$, $S_{yy}$, and $S_{xy}$ are the sample variance of $x$, the sample variance of $y$, and the sample covariance between $x$ and $y$. The estimator described in (2) is also called Deming regression (Deming, 1943). However, (2) is complicated by the value $\lambda = \frac{\sigma_\varepsilon}{\sigma_\delta}$, where $\sigma_\varepsilon$ and $\sigma_\delta$ are the variances of the random variables $\varepsilon$ and $\delta$, and these variances are generally unknown.

In practical applications, certain assumptions or approaches regarding $\lambda$ are employed to simplify (2). The most widely adopted approach is to set $\lambda = \infty$ ($\sigma_\delta = 0$), in which $x$ has only the deterministic part. As a result, this is the ordinary least squares regression (OLSR), where $\hat{b}_1 = \frac{S_{xy}}{S_{xx}}$. If $\lambda = 0$ ($\sigma_\varepsilon = 0$) is taken, we obtain another OLSR that treats $x$ as the dependent variable and $y$ as the independent variable, where $\hat{b}_1 = \frac{S_{yy}}{S_{xy}}$. For convenience, the former and the latter will be referred to as OLSR1 and OLSR2, respectively. The regression lines obtained from OLSR1 and OLSR2 typically differ, motivating the creation of a neutral slope. This is achieved by calculating the geometric mean of $\hat{b}_1$ derived from OLSR1 and OLSR2, yielding $\hat{b}_1 = sign(S_{xy})\sqrt{\frac{S_{yy}}{S_{xx}}}$, which is referred to as geometric mean regression (GMR). Alternatively, the GMR can be determined by assigning $\lambda$ as $\hat{\lambda} = \frac{S_{yy}}{S_{xx}}$ in (2). Finally, by assuming $\lambda = 1$, the orthogonal regression (OR) can be obtained as $\hat{b}_1 = \frac{S_{yy} - S_{xx} + \sqrt{(S_{yy} - S_{xx})^2 + 4 S_{xy}^2}}{2 S_{xy}}$. OLSR, OR, and GMR are commonly used to find a linear relationship between two datasets. In this application, the SST variations are designated $y$, and the time is assigned to $x$. Based on the overview provided, OLSR1 is likely to be the most suitable method within the linear regression family for capturing the long-term trend, especially considering that time lacks a random component. However, there has been a lack of careful examinations investigating the applicability of the regression family.

STL is a robust iterative nonparametric regression employing the loess smoother, facilitating the decomposition of a given time series into its long-term, seasonal, and remainder components, such as equation (1). Like other nonparametric regression approaches, STL requires the subjective selection of a smoothing parameter to delineate the lowest frequency component. STL comprises a sequence of intricate operations, which are divided into an inner loop and an outer loop, resulting in considerably longer computation times compared to the linear regression methods. Furthermore, the STL can also process the nonlinear relationship between $x$ and $y$. While the specifics regarding the implementation of STL are not addressed in this context, detailed information is reported in Cleveland et al. (1990). We thus assess the applicability of OLSR, OR, GMR, and STL for accurate temperature modeling.

## 3 Regression Analysis of Realistic Data

Three sets of SST data, collected from three coastal buoys located at Chenggong, Linshan Cape, and Magong stations (Fig. 1), all maintained by Taiwan's Central Weather Administration (CWA), were employed to assess the effectiveness of linear regressions and the STL. The Chenggong, Linshan Cape, and Magong stations are located on the eastern coast of Taiwan, the

northern coast of Taiwan, and the coast of Penghu Island, respectively. SST's hourly time series data from the three stations
(Figs. 2a, 3a, and 3b) exhibits temperature fluctuations of 8 °C to 15 °C, related to the seasonal variations, over its 13-year
measurement period. The variations encompass a consistent trend, overlaid with a seasonal cycle of approximately a year and
short-term fluctuations driven by tides and other oceanic processes. We summarize three distinct features of the SST time
series: (a) the time ($x$) lacks a random component, (b) the time ($x$) covers a significantly broader range than the SST ($y$), and
(c) the SST exhibits vigorous seasonal variations with amplitudes exceeding the magnitude of the long-term SST increase. All
the above features significantly affect the applicability of the regression family.

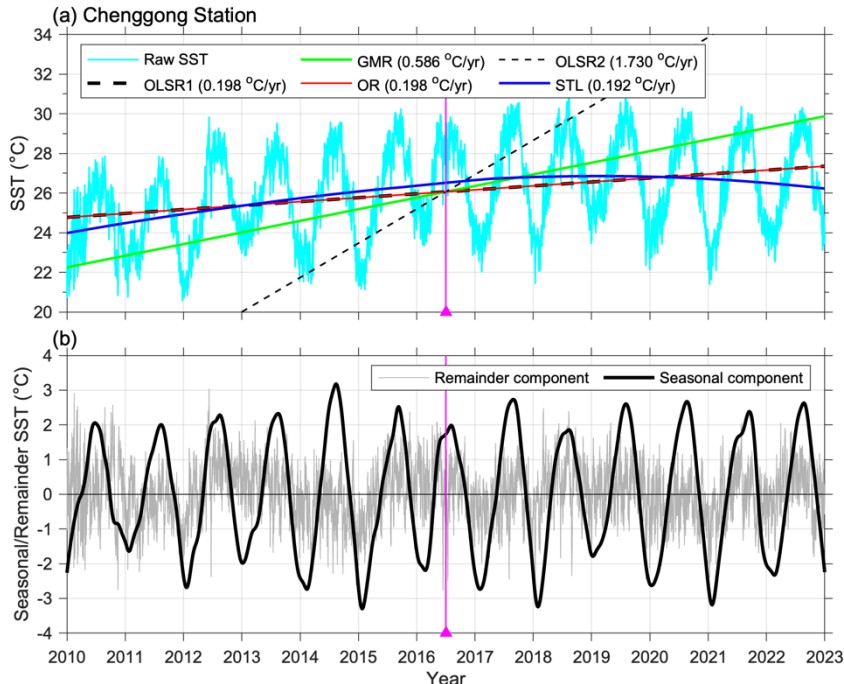

**Figure 2: (a) Time series (2010-2023) of sea surface temperature from Chenggong coastal buoy stations and its long-term trend estimated using OLSR1, OLSR2, GMR, OR, and STL methods. (b) The seasonal and remainder components of the STL result. The vertical magenta lines and triangles denote the mean value of the time axis.**

As depicted in Figs. 2a, 3a, and 3b, the long-term trends derived from the five methods are diverse, underscoring the necessity
for a thorough investigation into their applicability for accurate temperature modeling. Using the SST time series data from
the Chenggong station (Fig. 2a) as an illustration, the long-term trends estimated by the OLSR1 (thick black dashed line in
Fig. 2a) and OR (red line) methods are highly similar, revealing an SST increase of 0.198 °C/yr. In contrast, the GMR method
(green line) yields a different long-term trend than the previous two methods, indicating a rapid SST increase of 0.586 °C/yr.
Moreover, the OLSR2 method (thin black dashed line in Fig. 2a) yields an SST increase of 1.73 °C/yr. The outcome obtained
with the GMR and OLSR2 method is clearly unsatisfactory due to the excessively high estimated rate. The failure of OLSR2
is inevitable due to its reliance on the assumption that time ($x$) has a random component and SST ($y$) lacks such randomness.
This assumption directly contradicts how to properly model the dataset. As a result, the GMR method is also inappropriate

because its slope involves taking the geometric mean of $\widehat{b_1}$ derived from OLSR1 and OLSR2, causing its regression line (green line in Figs. 2a and Fig. 3) to fall between those of the two methods. Similar conditions hold true for the SST time series at the other two stations (Fig. 3). Thus, the OLSR2 and GMR methods will be excluded from our subsequent analysis. These results are summarized in Table 1.

**Table 1: Summary of the $\widehat{b_1}$ (unit: °C/yr) estimated using general linear regression, STL, evenized SST, SST anomaly, and a combination of linear and sinusoidal fitting. The slope derived from linear fitting to the STL nonlinear curve (blue lines in Figures 2a, 3a, and 3b) represents the $\widehat{b_1}$ value of STL. As for the methods of evenized SST, SST anomaly, and combined linear and sinusoidal fitting, the representative $\widehat{b_1}$ is determined as the mean value during its stable period, marked by the black dashed lines in Figure 7 (6 months trimmed time).**

| | Methods of general linear regression | | | | STL | Method of evenized SST | Method of SST anomaly | Method of linear and sinusoidal fitting |
| --- | --- | --- | --- | --- | --- | --- | --- | --- |
| | OLSR2 | GMR | OLSR1 | OR | | | | |
| Chenggong | 1.730 | 0.586 | 0.198 | 0.198 | 0.192 | 0.193 | 0.189 | 0.180 |
| Linshan Cape | 10.656 | 1.231 | 0.142 | 0.142 | 0.13 | 0.124 | 0.109 | 0.109 |
| Magong | 11.437 | 1.111 | 0.108 | 0.108 | 0.087 | 0.09 | 0.080 | 0.082 |

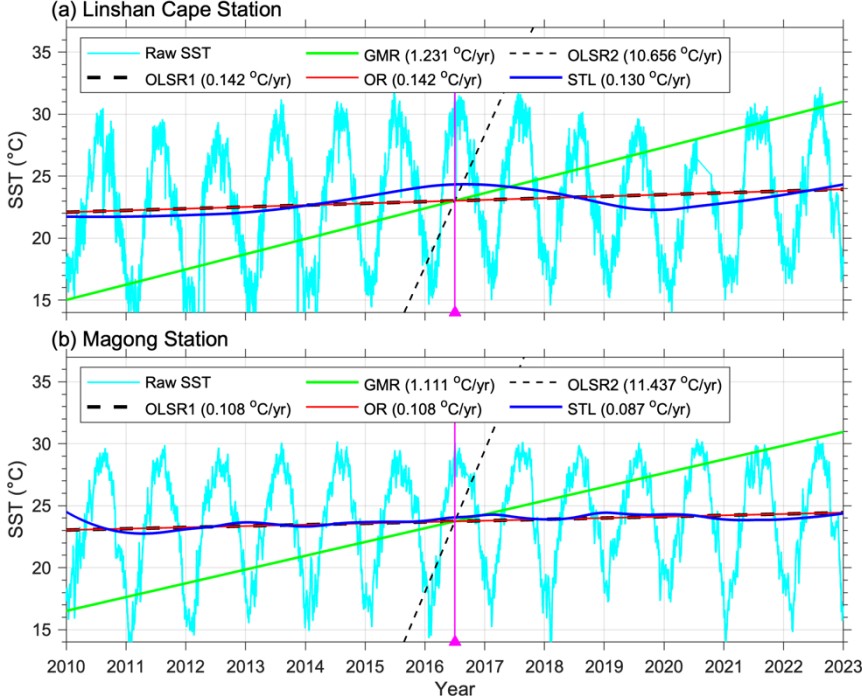

**Figure 3: Time series (2010-2023) of sea surface temperature from (a) Linshan Cape Station and (b) Magong Station and their long-term trend estimated using OLSR1, OLSR2, GMR, OR, and STL methods. The vertical magenta lines and triangles denote the mean value of the time axis.**

We next consider why OLSR1 and OR yield the same result despite having different estimators. Toward this explanation, we consider the geometric distance between each data point and the regression line. The slope of OLSR1 is determined by minimizing the sum of the squares of the vertical distances ($D_{OLSR1}$ in Fig. 4a) between the observed points and the assumed regression line. However, OLSR2 employs a similar approach but minimizes the sum of the squares of the horizontal distances ($D_{OLSR2}$ in Fig. 4a). The method of OR minimizes the sum of squares of the orthogonal distances ($D_{OR}$ in Fig. 4a) from the data points to the assumed regression line. Fig. 4a also contrasts two regression lines with larger and smaller slopes. When the regression line is nearly flat (regression line 2), $D_{OR}$ is very close to $D_{OLSR1}$. The orthogonal and vertical distances are very similar when the regression line is nearly horizontal (close to flat); therefore, the OR will closely approximate the OLSR1 regression. This set of conditions is generally true for long-term measured data. In estimating the regression lines depicted in Fig. 2a, we consider the time unit as a day. Accordingly, the time span is 4745 days, while the SST spans 10 °C, indicating that the slope would be nearly flat and that the lines of OR and OLSR1 are almost overlaid. The subsequent question is whether OR can consistently match OLSR1. It is well established that OR is not scale invariant. Changing the units of the variables will result in different regression lines. We test this effect by changing the unit of time to month and year. These changes do not affect the estimate when using OLSR1. Thus, in Fig. 4b, only the result of OLSR1 using the time unit of the day is plotted as a representative (thick black dashed line). In contrast, the regression line from OR, utilizing the time unit of the month, results in an estimated slope that is 0.1% larger (blue line in Fig. 4b), which may still be deemed acceptable. If the unit of year is used, implying that the number of 13 (years) is comparable to the SST variations of 10 (°C), the estimated SST increase rate is 0.278 °C/yr (magenta line in Fig. 4b). This is approximately 40% larger than that estimated using OLSR1, a difference that should be deemed unacceptable.

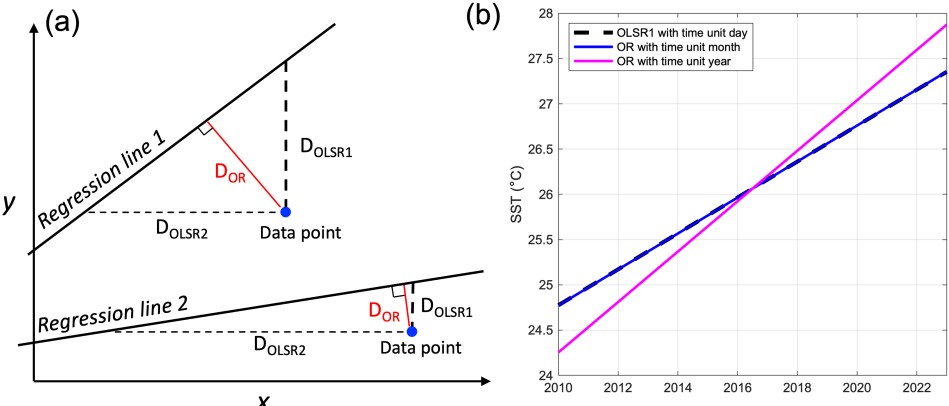

**Figure 4: (a) Schematic diagram showing the distances between the data point and the regression line used for OLSR1, OLSR2, and OR. Regression lines 1 and 2 represent steep and gentle slopes, respectively. (b) Regression lines derived from OLSR1 and OR with month and year time units using SST data collected at Chenggong station.**

Next, we proceed to the investigation of whether OLSR1 and OR genuinely capture the proper regression line. The long-term trend of Chenggong SST extracted by the STL, which may be the most suitable method (Tseng et al., 2010; Nidheesh et al., 2013), is illustrated as the blue curve in Fig. 2a. This curve exhibits a nonlinear trend characterized by a gradual increase from

2010 to mid-2019, followed by a decline until 2023. The 2-4 °C seasonal variation amplitudes are also well captured by STL, as the black curve in Fig. 2b indicates. The remainder of the curve encompasses the high-frequency tidal fluctuations and interannual variability (grey curve in Fig. 2b). The STL long-term curve (blue curve in Figure 3a) obtained using the SST data at Linshan Cape also exhibits a nonlinear trend. In the above two cases, although OLSR1 and OR cannot capture the nonlinearity of the long-term variations, they align neutrally with the STL curves. At Magong station, the nonlinearity of the curve (blue curve in Figure 3b) becomes weaker, allowing for better alignment between the OLSR1/OR lines and the STL curve. Finally, it remains imperative to demonstrate that OLSR1 and STL effectively capture the "true" regression line, necessitating using simulated data for validation.

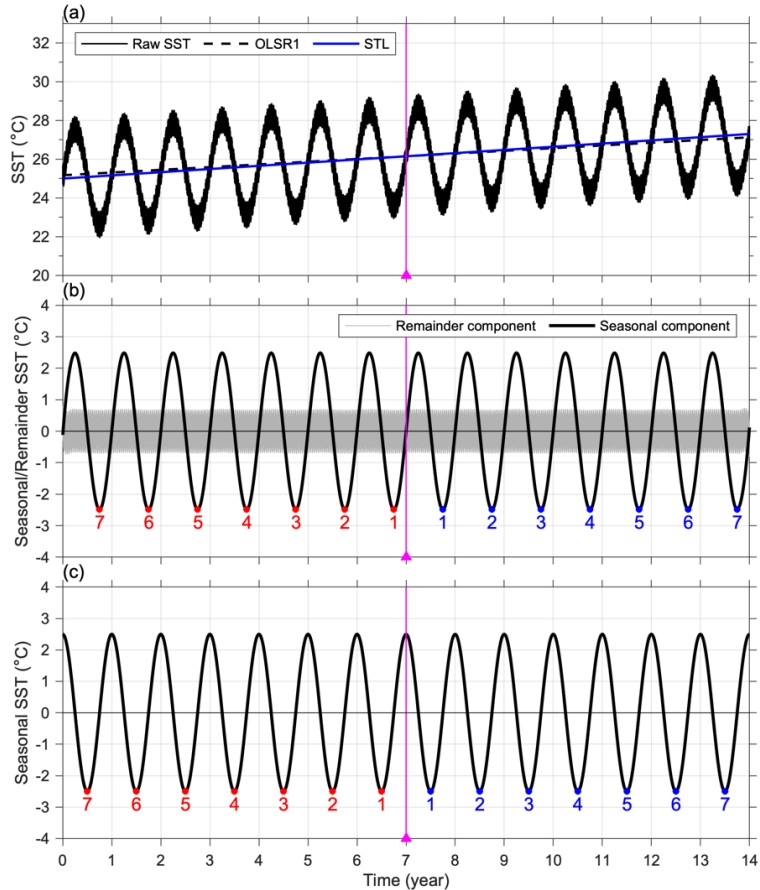

Figure 5: (a) Time series of simulated SST using (1) and (3) and its regressions using OLSR1 and STL, (b) seasonal and remainder components derived from STL, and (c) the seasonal variations using the cosine function shown as (4). The numbers in (b) and (c) denote the corresponding seasonal peaks on the two sides of the mid-time (magenta lines).

## 4 Examination using Simulated Data

Following (1), we first generate the test SST data: $ST = ST_{LT} + ST_{SV} + ST_R$, spanning 14 years (black curve in Fig. 5). $ST_{LT}$ is the long-term trend given by $ST_{LT} = 25 + qt$, where $t$ is the time with a unit of hour, and $q$ is the rate of SST increase. Here, $q$

=1.8754×10⁻⁵ °C/hr is given, equivalent to 0.1643 °C/yr. This equation signifies a linear temperature increase of 2.3 °C over the 14 years. The seasonal variation is represented as a sine function

$$ST_{SV} = A_{SV} \sin (2\pi t/T_{SV}), \tag{3}$$

where $A_{SV}$ and $T_{SV}$ are 2.5 °C and 365 days, respectively. The remainder component ($ST_R$) incorporates diurnal (24-hour period) and semidiurnal (12.42-hour period) tidal variations, each of which has an amplitude of 0.5 and 0.2 °C, respectively. The regression line of the test dataset (blue line in Fig. 5a), derived from STL, precisely overlays its long-term trend $ST_{LT}$ that is not plotted because it would be covered. In addition, the seasonal variations and the remaining components are effectively decomposed, as shown in Fig. 5b. OLSR1 underestimates the slope at 0.14 °C/yr, which is 15% lower than the actual value (black dashed line in Fig. 5a). The obtained result is unexpected, considering the good correspondence observed between STL and OLSR1 in the realistic data (Figs. 2a, 3a, and 3b). We further clarify the mechanism behind this underestimation by replacing the $ST_{SV}$ as a cosine function

$$ST_{SV} = A_{SV} \cos (2\pi t/T_{SV}) \tag{4}$$

in the test data. As a result, OLSR1 accurately predict the slope, indicating that the phase of the seasonal cycle holds significance in this context.

The linear regression line must pass through the point of sample mean of $t$ and $ST$ ($\bar{t}, \overline{ST}$). Therefore, we can model the regression line as a lever with a fulcrum at ($\bar{t}, \overline{ST}$), adjusting its angle (or slope) to ensure that positive residuals entirely cancel out negative residuals. Similar to the concept of torque, the greater the distance of a deviated data point from $\bar{t}$, the more substantial its impact on adjusting the line's angle. In our original test data (equation (3)), the seasonal variation is modeled as a sine function, an odd function with respect to $\bar{t} = 7$ (Fig. 5b). The high-leverage points arising from seasonally induced deviations (black line in Fig. 5a) from the regression line play a role in determining the slope of the line. This effect can be further illustrated by examining the seasonal variation depicted in Fig. 5b, specifically focusing on the seasonal peaks. We consider the 14 peaks at $ST_{SV}$ <0, which are labelled as 1-7 in blue and red on the right-hand and left-hand sides of the fulcrum ($\bar{t} = 7$; denoted as magenta lines), respectively. The 7 peaks on the right-hand side favor a clockwise rotation of the line. Consequently, they tend to decrease the slope. In contrast, the 7 peaks on the left-hand side favor a counterclockwise rotation of the line, contributing to an increase in the slope. However, the two opposing tendencies on the right and left-hand sides are unbalanced. Peak #1 on the right-hand side is $\frac{3}{4}T_{SV}$ from the fulcrum, and its corresponding peak #1 on the left-hand side is $\frac{1}{4}T_{SV}$ from the fulcrum. Overall, the seasonal peaks on the right-hand side consistently have a longer distance of $\frac{1}{2}T_{SV}$ than their counterparts on the left-hand side. Based on the concept of torque, this indicates that the overall effect of the seasonal peaks at $ST_{SV}$ <0 is to lower the slope. The peaks at $ST_{SV}$ > 0 also act to lower the slope if the same treatment is applied. Therefore, we can conclude that the net effect of the 14 seasonal peaks is to lower the slope, so that the long-term trend is underestimated. We term this as phase-distance imbalance.

The bias of phase-distance imbalance does not manifest in the case of an even function, such as a cosine function (equation (4) and Fig. 5c). The resulting data show that the new $ST_{SV}$ appears to be a mirror pattern with respect to $\bar{t} = 7$, which is a typical

feature of an even function. Clearly, the peak pairs distributed on both sides of the fulcrum at $\bar{t} = 7$ have identical distances to the fulcrum. Our previous analysis has pointed out that the seasonal variations do not introduce bias in estimating its long-term trend. It is suggestive that addressing the bias induced by phase-distance imbalance may involve trimming the data to

ensure that $ST_{SV}(t - \bar{t})$ within the dataset becomes an even function.

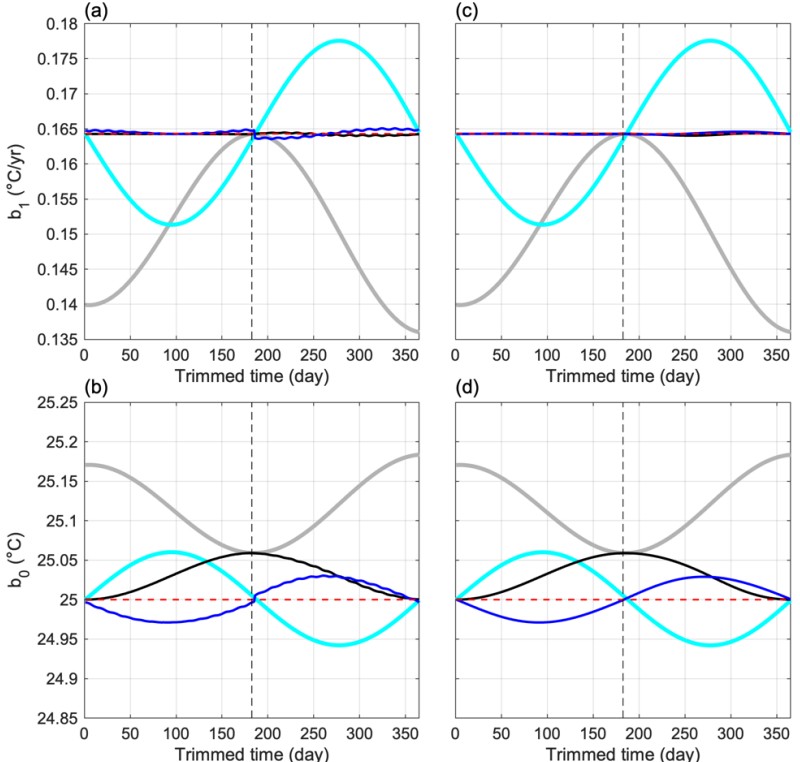

**Figure 6: Relationship (a) between the trimming time and $\widehat{b_1}$ and (b) between the trimming time and $\widehat{b_0}$ for simulated SST (1) with $ST_{SV}$ represented as a sine function (3) (grey curve) and a cosine (4) (cyan curve). (c) and (d) are the same as (a) and (b), respectively, but removing the tidal signals $ST_R$. The black and blue curves depict the outcomes after correcting the seasonal bias through steps**
**1-7.**

This task involves trimming the test data (black curve in Fig. 5a) for the last $N$ days ($N$ = 0, 1, 2... 365) and subsequently conducting OLSR1 to obtain $\widehat{b_1}$ (grey curve in Fig. 6a) and $\widehat{b_0}$ (black curve in Fig. 6b). When the trimmed time is 0 days, $\widehat{b_1}$=0.14 °C/yr and $\widehat{b_0}$=25 °C, which represent the initial regression line, shown as the dashed black line in Fig. 5a. As the number of trimmed days increases, $\widehat{b_1}$ (black curve in Fig. 5a) gradually rises toward the correct value of 0.1643 °C/yr (as

marked by the grey dashed line) at trimmed time = 182.5 days ($\frac{1}{2}T_{SV}$), before subsequently decreases again. It is worth noting that $ST_{SV}(t - \bar{t})$ becomes an even function when the data in the last $\frac{1}{2}T_{SV}$ are trimmed off (Fig. 5b), supporting for our proposed concept.

The value of $\widehat{b_0}$ still deviates from the correct value of 25 °C at a trimmed time = 182.5 days ($\frac{1}{2}T_{SV}$) (grey curve in Fig. 6b). This deviation is attributed to the change in $\overline{ST}$ that is induced by trimming the data, which can be corrected by vertically

displacing the regression line to intercept with the original $\overline{ST}$. Another example illustrates the case of an even (cosine) function in the initial stage (equation (4)). The trimming of the data causes the initially correct $\widehat{b_1}$ and $\widehat{b_0}$ to deviate, except for trimmed times of $\frac{1}{2}T_{SV}$ and $T_{SV}$. This example also demonstrates that both $\widehat{b_1}$ and $\widehat{b_0}$ can be underestimated or overestimated. We revisited the realistic data collected from the CWA's three stations, applying the insights obtained from examining the simulated data. The seasonal cycle regularly exhibits a higher SST in August and a lower value in February (Figs. 2 and 3).

To make the SST an even function, it is desired that mid-time ($\bar{t}$) is located at the maximum or minimum SST. The regression lines derived from our three SST datasets using OLSR1 align reasonably well with the STL curves because the datasets tend to approach an even function, although not precisely. Highlighting the mid-time line (magenta) in Figs. 2 and 3 can illustrate this. The mid-time falls in June 2016, just 2 months away from the peak SST in August 2016. Improving the slope estimation accuracy might involve data trimming to position the mid-time of the dataset precisely in August 2016. Indeed, Figures 6a and

6b emphasize a significant uncertainty in estimating $\widehat{b_1}$ and $\widehat{b_0}$, depending upon the seasonal phases and magnitudes. Consequently, having a longer dataset does not necessarily guarantee a more accurate outcome.

## 5 Implementation in correcting the seasonal bias

### 5.1 Methodology

To correct the seasonal bias, we consider a pre-processing procedure that trims the data to ensure an even function before running the regression analysis. For convenience, we term this procedure "evenization", following an analogous concept introduced in signal processing (Kahn, 1957; Kondo and Kou, 2001). This could involve removing the initial portion of the data until the remaining set reflects an even pattern around its mid-time. This adjustment helps reduce or eliminate the bias introduced by the seasonal patterns. Numerous methods are expected to achieve this correction. Here, we propose a procedure

as follows:

**Step 1**: *Detrending*. The long-term trend is derived by applying the OLSR1 to the raw SST data. A detrended time series *DTR_DATA* is obtained by subtracting the long-term trend from the raw SST data.

**Step 2**: *Data Folding*. *DTR_DATA* is split evenly at its midpoint, forming two segments of equal length: the first half segment, *D1*, and the "flipped" second half segment, *D2*.

**Step 3**: *Hilbert Transform*. Convert *D1* into analytic signal $A_{D1}$, a complex-valued function comprising the real part *D1* and the imaginary part $\widehat{D1}$. $\widehat{D1}$ is a $\pi/2$ shifted function of *D1* that can be obtained via the Hilbert transform (Marple, 1999) of *D1*. $A_{D2}$, the analytic signal for *D2*, can be derived similarly.

**Step 4**: *Complex Correlation*. Computing the correlation coefficient between $A_{D1}$ and $A_{D2}$ yields a complex correlation, having magnitude $C$ representing the maximum correlation and phase angle $\theta$ where the maximum correlation occurs. $\theta$ typically falls between $-\pi$ and $\pi$. $\theta$ represents the phase by which *D1* leads *D2*.

**Step 5**: *Trimming Data*. If $\theta \geq 0$, the raw SST data in the period of the initial $T_{SV} - \frac{\theta}{2\pi} T_{SV}$ is trimmed off. If $\theta < 0$, the raw SST data in the period of the initial $\frac{|\theta|}{2\pi} T_{SV}$ is trimmed off. These ensure that the remaining data approximates an even function by trimming to the minimum dataset length, which is shorter than $\pi/2$ (3 months).

**Step 6**: *Obtaining the slope $\widehat{b_1}$*. The long-term slope is estimated by applying OLSR1 to the trimmed SST dataset.

**Step 7**: *Obtaining the intercept $\widehat{b_0}$*. Data trimming induces change of $(\bar{t}, \overline{ST})$. Therefore, the intercept is obtained by ensuring the new regression line pass through original $(\bar{t}, \overline{ST})$. Returning to Step 1 for further detrending using the derived long-term trend is an optional step that could be carried out iteratively. However, this additional iteration did not significantly impact the results in our cases. Nevertheless, this option remains open for datasets that may benefit from further iterations.

The application of our procedure steps 1-7 to the simulated datasets, comprising a sine seasonal cycle and a cosine seasonal cycle, is shown as the black and blue lines in Figs 6a and 6b, respectively. Our corrected $\widehat{b_1}$ align closely with the known $b_1$ value marked as the red dashed line (Fig. 6a). The seasonal bias-related uncertainty, initially at O(10$^{-2}$) °C/yr, has been notably reduced to O(10$^{-3}$) °C/yr. The wiggles and discontinuity observed around the trimmed time of 184 days, contributing to an uncertainty at O(10$^{-3}$) °C/yr, stem from tidal influences. This is evidenced by re-examining the dataset (1) after removing the tidal component, $ST_R$, showing the absence of the wiggles and discontinuity (Fig. 6c). These tidal signals might slightly impact the determination of $\theta$ in step 4 of the process. The corrected estimate of $\widehat{b_0}$ converges closely to the known $b_0$ value of 25 °C (depicted as the red dashed line), exhibiting a deviation smaller than 0.07 °C (Fig. 6b). Similarly, the tidal effect could produce small wiggles but can be corrected by excluding the tidal component in the analysis (Fig. 6d).

Our approach is subsequently employed with the SST datasets collected by CWA (as shown in Fig. 7). While the true $\widehat{b_1}$ remains unknown within realistic data, we can assess their coherence through data trimming from the end. It's generally understood that the long-term trend of a 10-year or longer time series doesn't significantly change when its length decreases by six months. At the Magong station, the estimated range of $\widehat{b_1}$ spans from 0.07 to 0.13 °C/yr when employing the general OLSR (grey curve in Fig. 7a). In contrast, our approach significantly reduces uncertainty to a range of 0.08 to 0.1 °C/yr (red dots in Fig. 7a) within the initial six-month trimmed time. The consistency of $\widehat{b_1}$ can reach ten-month trimmed time at the Linshan Cape (Fig. 7b) and Chenggong (Fig. 7c) stations. These findings provide support for the effectiveness of our proposed method. To maintain the nature of the long-term trend behind the data, the trimmed data length ought not to exceed a seasonal cycle. The representative $\widehat{b_1}$ could be the mean value during its stable period. The recommended length of the stable period would be half of a seasonal cycle, i.e., six months. As a result, the representative $\widehat{b_1}$ values at Magong, Linshan Cape and

Chenggong stations are 0.09 °C/yr, 0.124 °C/yr, and 0.193 °C/yr (Table 1), respectively, as marked by the black lines in Fig. 7.

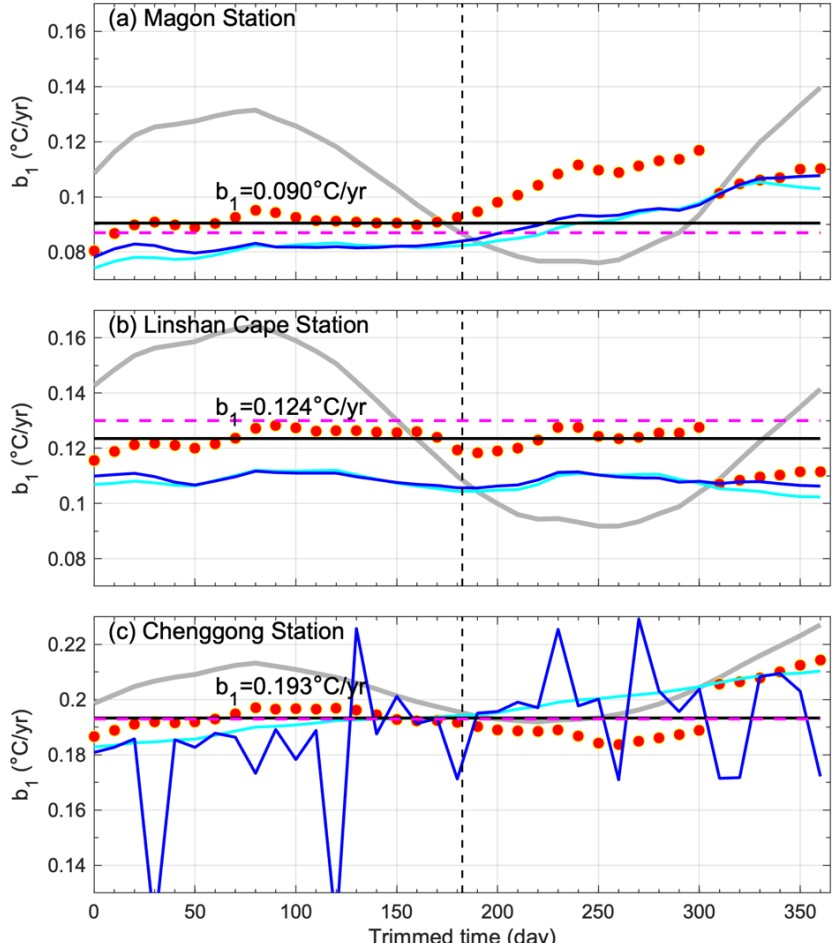

**Figure 7: $\widehat{b_1}$ as function of trimmed time using the SST data collected in (a) Magong, (b) Linshan Cape, and (c) Chenggong stations. The OSLR1 method and our proposed method (steps 1-7) are represented respectively by the grey curves and red dots. The black lines depict the averaged $\widehat{b_1}$ values based on corrected data within a trimmed time shorter than $T_{sv}/2$, indicated by the vertical dashed line. The cyan and blue curves are the slope estimated using SST anomalies and combined linear and sinusoidal fitting methods, respectively. The magenta dashed lines are the slope derived from linear fitting to the STL nonlinear curve.**

## 5.2 Comparison with conventional methods

We have demonstrated our proposed evenization method as a feasible approach to estimate the long-term trend of SST. It is desired to compare our method with the commonly used methods in the climate community. The first method involves computing the daily climatological value of SST using the available SST data. The long-term trend can be estimated by applying OLSR1 to the SST anomalies derived by subtracting the climatological SST from the original SST data. This is

expected to lower the bias resulting from seasonality. The second method models the SST data as a combination of linear and sinusoidal functions (e.g., Park et al., 2022):

$$SST(t) = b_0 + b_1 t + A\cos\left(\frac{2\pi t}{T}\right) + B\sin\left(\frac{2\pi t}{T}\right). \qquad (5)$$


The first two terms on the right-hand side are the linear function, representing the long-term trend. The third and fourth terms on the right-hand side represent the seasonal component, where $T$=365 days is the period. The amplitude of seasonal variations can be obtained as $\sqrt{A^2 + B^2}$. Here, $\widehat{b_o}$, $\widehat{b_1}$, $\hat{A}$, and $\hat{B}$ are obtained using nonlinear least squares fitting the SST dataset. Adding other periodic components, such as interannual variations, may only sometimes be helpful due to the increased number of

fitting parameters, which could lower the numerical accuracy and stability.

The performance of the three methods is evaluated using the 14-year time series, as depicted in Fig. 5a, which is generated using equation (3). The semidiurnal tidal amplitude is increased from 0.2 °C to 0.3 °C to better investigate the impacts of small fluctuations. Fig. 8 shows how the estimated slope changes with different data lengths (3-14 years) used for estimation, allowing for the evaluation of the uncertainty of each method. Overall, as the data length increases, there is a reduction in $\widehat{b_1}$

uncertainty for both the linear trends of SST anomalies (cyan curve in Fig. 8a) and evenized SST (red curve). In both methods, the uncertainty of $\widehat{b_1}$ is significant when the data length is less than seven years, and the deviation could reach 0.01 °C/yr. The deviation is less than 0.003 °C/yr for data length > 7 years. $\widehat{b_1}$ obtained from the evenized SST method closely aligns with the correct value (represented by the black dashed line), whereas $\widehat{b_1}$ obtained from the SST anomalies method tends to be consistently lower than the correct value. This can be clearly found in the probability density function (PDF) shown in Fig. 8b.

The PDF of $\widehat{b_1}$ estimated using the evenized SST is unbiased because it concentrates around the correct value (0.1643 °C/yr; the vertical dashed line). In contrast, the SST anomalies estimate (cyan line) is biased because its PDF deviates from the correct value. This suggests our method could be a better estimator.

When using the combined linear and sinusoidal fitting method, there is no clear relationship between the uncertainty and data length, as depicted in Fig. 8a by the blue line. The PDF shows a more concentrated distribution at $\widehat{b_1}$=0.1643 °C/yr (blue line

in Fig. 8b), suggesting better performance than our method. Figure 9a shows a successful fitting curve of (5) (blue solid line), which overlaps with the simulated data (red solid line), when the data length used is 8 years. The resulting long-term trend (blue dashed line) also aligns with that from the evenized SST (red dashed line, but it is exactly covered by the blue dashed line). However, unexpected fitting failures can cause large deviations (blue line in Fig. 8a), such as in the example when the data length of 7 years is used (Fig. 9b). The fitting curve (blue solid line in Fig. 9b) has a smaller seasonal amplitude and a

clear phase shift compared to the simulated data (red solid line in Fig. 9b). The estimated slope of the long-term trend (blue dashed line) is gentler than the known trend. In contrast, the known trend agrees with that estimated using evenized SST (red dashed line). We re-examine the PDF using $\widehat{b_1}$ with data length > 7 years, which is generally applicable for long-term trend estimates (Fig. 8c). The method of SST anomalies remains biased. The methods of linear and sinusoidal fitting are unbiased, and the peak value of PDF slightly increases from 0.79 to 0.81. Similarly, the methods of evenized SST are unbiased, but the

peak value of PDF significantly increases from 0.32 to 0.5. To summarize, our proposed method is unbiased and better than

the conventional SST anomalies method. While our method may have a more significant degree of uncertainty than linear and sinusoidal fitting, this uncertainty remains within an acceptable range. Furthermore, linear and sinusoidal fittings can be unstable when applied to natural data containing significant noise.

The same examination using the CWA's data for SST anomalies (cyan lines in Fig.7) and combined linear and sinusoidal fitting (blue lines in Fig. 7) methods is carried out. Again, we focus on the comparison in the stable period, six months trimmed time. $\hat{b}_1$ obtained using SST anomalies is 0.004-0.015 °C/yr lower than obtained using evenized SST (cyan lines in Fig. 7 and Table 1). The obtained result agrees with the expected outcome based on the simulated data, as shown in Fig. 8. Using the combined linear and sinusoidal fitting (blue lines), the obtained $\hat{b}_1$ roughly aligns with SST anomalies. The result differs from the simulated data. It is anticipated that complex and diverse natural signals could have interfered with the fitting results, often considered noise. Indeed, unexpected peaks related to the failed fitting occur for the data in the Chenggong station (Fig. 7c), where the tidal signal (Fig. 2a) is strongest among the three stations. Finally, the slope obtained from linear fitting to the STL nonlinear curve (magenta dashed lines in Fig. 7 and Table 1) is close to the result obtained from evenized SST.

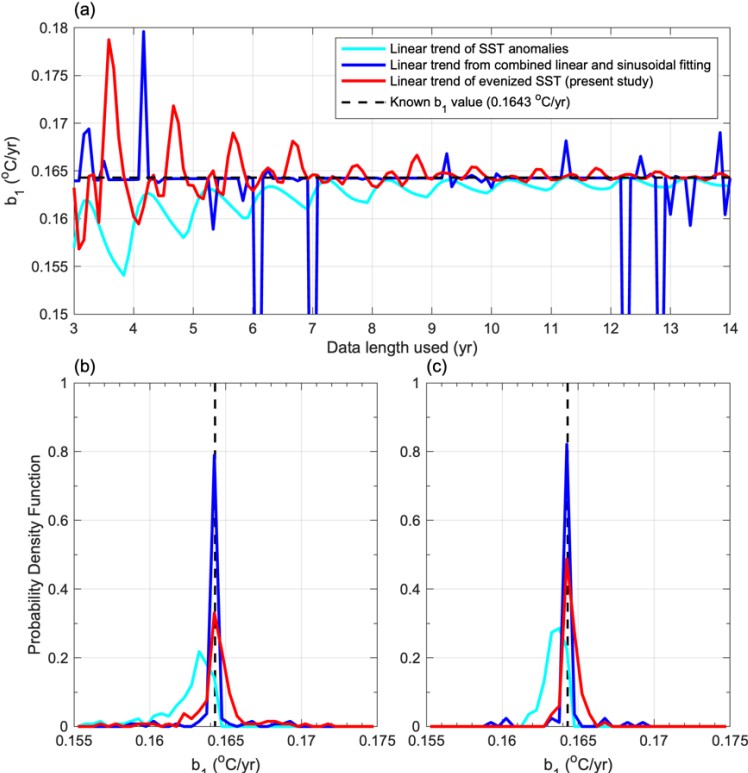

**Figure 8: (a)** $\widehat{b_1}$ **as function of data length using the simulated SST data and probability density function of** $\widehat{b_1}$ **using data length of (b) 3-14 years and (c) 7-14 years by applying the methods of SST anomalies (cyan lines), combined linear and sinusoidal fitting (blue lines), and evenized SST (red lines). The black dashed lines denote the known b₁ value.**

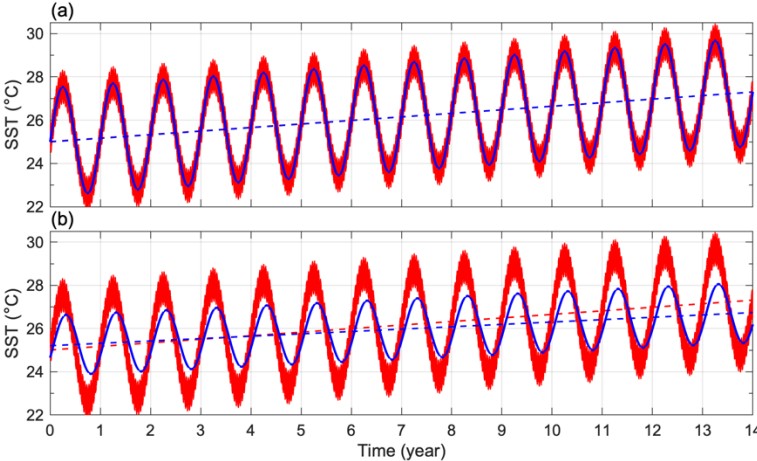

**Figure 9: Linear and sinusoidal fitting curve of equation (5) using simulated data lengths of (a) 8 years and (b) 7 years. The red and blue solid lines represent the simulated data and its fitting curve, respectively. The red and blue dashed lines represent the long-term trends from the evenized SST and fitting methods, respectively.**

## 6 Discussion and Conclusions

Here, we systematically examine the methods of linear regression and STL for their application to extract the long-term trend from an SST time series. The STL may be the best method to extract the long-term trend accurately. However, it comes with a significantly long computational time. The runs conducted here using idealized data require approximately 3-5 minutes each. The linear regression methods are usually shorter than 0.1 s. STL is not well suited for tasks that involve lengthy loop operations, e.g., computing the global increase rate of SST using satellite data. Instead, linear regression methods are preferable for such tasks but are subject to the nature of the long-term SST data. We summarize three distinct features of the SST time series: (a) the time-axis lacks a random component, (b) the time covers a significantly broader range than the SST, and (c) the SST exhibits vigorous seasonal variations with amplitudes exceeding the magnitude of the long-term SST increase. Feature (a) indicates that the most suitable linear regression method is OLSR, which considers SST as composed of a deterministic and a random part, while time has only a deterministic part. The alternative OLSR that swaps the consideration of SST and time is evidently unsuitable. The GMR is excluded because it also accounts for the randomness of time. The second feature (b) indicates that the slope is nearly flat because the time spans a large interval, e.g., 3000 days, while the SST range is typically smaller than 10 °C. When the regression line is nearly horizontal, the OR will closely approximate the OLSR. This is generally true for long-term measured data.

Accordingly, we propose that OLSR (and OR) can be employed to extract the long-term trend of SST data by addressing the bias arising from feature (c). The proper regression to obtain the long-term trend depends on the effective cancellation of the strong seasonal signals. However, effective cancellation only occurs when the seasonal signal is an even function, indicating that it has a mirroring trend with respect to the mid-time. The bias will be the strongest when the seasonal signal is an odd function. We refer to this temporal phenomenon as "phase-distance imbalance." This bias induced by the seasonal signal could

be appropriately corrected by trimming the data, ensuring that the dataset becomes an even function before conducting OLSR (or OR). Finally, we compare our methods with two commonly used methods in the climate community. Our proposed method is unbiased and better than the conventional SST anomalies method. While our method may have a more significant degree of uncertainty than linear and sinusoidal fitting, this uncertainty remains within an acceptable range. The fitting method generally performs better, as seen in Figure 8. However, linear and sinusoidal fittings can be unstable in occasional cases. The poor fitting may be addressed by providing better initial guess values, constraining parameter intervals, changing the numerical method, filtering the data, and other approaches. All of these require additional trials. Our proposed method provides another robust and efficient method that can avoid this disadvantage. Users can choose the method that best suits their analysis needs.

### Code availability

The MATLAB code for linear regressions and seasonal bias correction is available https://doi.org/10.5281/zenodo.10360553.

### Data availability

The experimental SST data is available at https://doi.org/10.5281/zenodo.10360553.

### Author contribution

MHC and YHC initiated and formulated the study's conceptual framework. MHC and YCH contributed to developing the methods to correct the seasonal bias, conducted statistical analyses, and produced figures. MHC undertook the literature review and drafted the initial manuscript. CTT provided insight into the application of our method to the realistic data. YHC, CTT, JC, and JCJ contributed to the revision and editorial enhancement of the text.

### Competing interests

The authors declare that they have no conflict of interest.

### Acknowledgments

This paper's inception stems from our endeavor to estimate the long-term sea surface temperature trend concerning global warming around Taiwan. We thank Taiwan's Central Weather Administration (CWA) for funding this research (Grant ID: 1122057C). Discussion with Yu-Yu Yeh has been very helpful. We acknowledge the outstanding efforts of the personnel in CWA responsible for maintaining the coastal buoys over a long period, facilitating our study. We thank the editor and both reviewers for their constructive suggestions, which have greatly improved the original manuscript.

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
