# Peer review of "Revisiting regression methods for estimating long-term trends in sea surface temperature"

_Natural Hazards and Earth System Sciences, 2023_

## Referee Comment (RC2)

**Review of "Revisiting regression methods for estimating long-term trends in sea surface temperature" by Chang et al.**

This study is an interesting study that revisits a set of regression methods that can be used to estimate long-term trends in sea surface temperature. Authors methodically investigate these regression methods and identify their pros and cons for estimating trends in sea surface temperature. Overall, this work is important and I find that the work deserves to be published in NHESS and falls within the scope of NHESS. However, before publication, authors should consider multiple revisions that will improve the understanding of the manuscript and background knowledge of the work to readers. Therefore, I suggest the below revisions.

Decision – major revision

**Major Comments**

1. Literature Review

   L29-31: Authors highlight some background knowledge on how global warming has led to increased warming in the ocean and how warmed oceans can change circulation patterns and provide energy for tropical storms. However, they only mention one reference for this sentence - Lin and Chan (2015) which explains a recent decrease in typhoon destructive potential. Authors need to revisit their literature review and provide a set of references to back their claim on how warming oceans can modify circulation patterns as well as how they can contribute to intense and tropical storms.

   L32-33: Again, one reference is not enough to support the claim on how warming oceans can influence the marine environment.

   In both cases, there are many novel research has been conducted and they have to be credited properly when stating these claims.

2. Explanation of Buoy Data

   The authors state that they have used SST data collected from three coastal buoys. They should provide locations of these buoys using geographical coordinates. I strongly recommend including a map to present the general location of these buoys.

3. Figures

   For figures 1(a) and 2 authors should indicate the numerical values of slopes for each regression method. Preferably using the same colors of regression lines. This will help the readers to compare the regression estimates by examining figures without going back and forth through text and figures. For Raw SST, please use a different color from OLSR1.

4. Methods

In Figure 1(b), the authors plot the remainder components. However, it is unclear in the methods section how they obtain these remainder components. Please explain this in detail in the methods sections.

5. A summary table.

The authors do a good job of comparing the outcomes of different regression methods within the text of the manuscript. Since the comparison of these regression methods is the core objective of this paper, the authors should highlight their findings appropriately. For this, I suggest using a table to summarize and compare the findings. This will provide readers the opportunity to go through the findings of this important work at one stop rather than scouring through the text for each regression method.

**Minor Comments**

1. L59-61: I assume µ is mean. But it will be helpful to readers if you define it properly.
2. L105:    (2) should be (b)
3. L162:    Recheck figure reference (Fig 3 or 4?)
4. L259-261: It seems the letter "O" is used instead of the number zero (0).
5. L299: change "real-sea" to "observed"

---

## Author Comment (AC1)

Responses to Reviewer #1

*Thanks to the reviewer for the critical and very constructive comments. We've done our best to find the advantages of our developed methods by comparing them with the previous methods.*

In this manuscript, the authors compare the ability of several regression techniques to recover underlying long-term trends in daily SST (Sea Surface Temperature) records. They find that the ordinary least squares (OLS) method could be biased due to an imbalanced phase associated with seasonal cycles. They further propose a 7-step approach to account for imbalanced seasonal cycles to obtain a less-biased estimator. Despite the interesting technical discussion, the manuscript, in my view, does not provide sufficient methodological advances. The atmospheric and climate community already has other approaches to address this issue, potentially more efficiently (please refer to my second comment below).

*Thanks for the comments. As suggested, we've compared our method with two commonly used methods in the atmospheric and climate community in detail, which will be illustrated below.*

Moreover, I am concerned that the topic of this paper might not fall within the scope of this journal, which focuses on Natural Hazards. As a result, I would suggest that the authors compare their methods to the approach I suggest below and consider submitting this work to a more technical journal.

*We had extensively searched for an appropriate journal for this study. After careful consideration, we have identified NHESS as a suitable option, as it welcomes manuscripts focusing on methodological advancements that contribute to addressing natural hazards, including issues related to global warming. Hence, we are motivated to submit our work to NHESS.*

1. I appreciate the authors' careful introduction of OLS2, GMR, and OR, but this discussion may not be entirely relevant in this context because the timing of SST measurements should be well-known. The authors also point this out in line 115. On the other hand, the STL (Seasonal-Trend Decomposition procedure based on Loess) probably deserves a more detailed introduction, including the mathematics and equations.

*The details of STL have been described in Cleveland et al. (1990). The conceptual*

*description has been provided in L102-109. As the methodology of STL involves numerous trivial and detailed operations, we don't think it is suitable to incorporate into the present manuscript. We've cited the paper of Cleveland et al. (1990) for readers interested in the details.*

2. In atmospheric, ocean, and climate research, the first step of an analysis typically involves removing the seasonal cycle. Long-term trends are then estimated from the anomalies. When using daily data, directly calculating daily climatology often results in noisy estimates. Hence, the community uses sine and cosine functions to fit the amplitude and shape of seasonal cycles. For a problem that also estimates long-term trends, the model would be:

$T = \mu + k*yr + \sum_{i=1}^{N}(a\_i * sin(i*yr*2\pi)) + \sum_{i=1}^{N}(b\_i * cos(i*yr*2\pi))$,

where the goal is to fit for μ, k, a_i, and b_i from the data, with i = 1, 2, ..., N. In practice, this is simply a multi-linear regression, and N can be determined if increasing N does not further improve the fitting (using, for example, an F-test). Fitting sine and cosine functions simultaneously captures different phases. Hence, unless the authors demonstrate that their method outperforms the community's common practice, I am not fully convinced that the method they propose would make a significant methodological improvement.

*We must emphasize that we do not intend to replace the previous methods but rather propose another way to achieve the long-term trend estimate. After adding a sub-session to compare our method with the previous two, we found that our method is robust and has advantages over the other two in certain aspects. Please see session 5.2 and Figures 7 and 8.*

[revised manuscript text omitted]

---

## Author Comment (AC2)

Responses to Reviewer #2

*We want to express our gratitude to the reviewer for the valuable and constructive comments. We have carefully considered each comment and greatly appreciate your input.*

This study is an interesting study that revisits a set of regression methods that can be used to estimate long-term trends in sea surface temperature. Authors methodically investigate these regression methods and identify their pros and cons for estimating trends in sea surface temperature. Overall, this work is important and I find that the work deserves to be published in NHESS and falls within the scope of NHESS.

*Thank you.*

However, before publication, authors should consider multiple revisions that will improve the understanding of the manuscript and background knowledge of the work to readers. Therefore, I suggest the below revisions.
Decision – major revision

*We've revised the manuscript accordingly. We have tried our best to introduce the background knowledge more concretely and clearly and added 16 papers as relevant references. Please see our illustrations in the following reply.*

Major Comments
1. Literature Review
L29-31: Authors highlight some background knowledge on how global warming has led to increased warming in the ocean and how warmed oceans can change circulation patterns and provide energy for tropical storms. However, they only mention one reference for this sentence - Lin and Chan (2015) which explains a recent decrease in typhoon destructive potential. Authors need to revisit their literature review and provide a set of references to back their claim on how warming oceans can modify circulation patterns as well as how they can contribute to intense and tropical storms.

*Thanks for the comments. As suggested, we've elaborated on the impacts of increased SST on typhoons and ocean circulation based on a series of papers cited in the revised manuscript. See L33-43.*
*"Rising sea temperatures have the potential to cause changes in ocean circulation*

*patterns. Research has shown that the Kuroshio and Gulf Stream, two important subtropical western boundary currents in the North Pacific and North Atlantic, can become stronger (Sakamoto et al., 2005; Cheon et al., 2012; Chen et al., 2019; Wang and Wu, 2019) and weaker (Levermann et al., 2005; Chen et al., 2019), respectively. This can ultimately impact the Atlantic meridional overturning circulation (AMOC), as the Gulf Stream is a key system component. The impact of SST warming on tropical cyclones has been a top concern in recent decades (Emanuel, 2005). As global warming continues, we see fewer cyclones overall, but those that do occur are more powerful, longer-lasting, larger, and more destructive (Emanuel, 2005; Maue et al., 2011; Lin et al., 2014; Sun et al., 2017). This increase in destructive potential is due to the combination of longer storm lifetimes and greater storm intensities resulting from warmer sea surface temperatures. However, the situation may be more nuanced, as other atmospheric conditions, such as increased wind shear, could counteract or even reverse this trend of heightened destruction (Lin and Chan, 2015)."*

L32-33: Again, one reference is not enough to support the claim on how warming oceans can influence the marine environment. In both cases, there are many novel research has been conducted and they have to be credited properly when stating these claims.

*Thanks. See L43-47.*
*"Coral reefs are facing an increasing threat due to rising ocean temperatures (Pandolfi et al., 2011). This has resulted in the unprecedented mass bleaching of corals, which has been triggered by rising sea surface temperatures (Frieler et al., 2013; Hughes et al., 2017; Hoegh-Guldberg et al., 2017; Sully et al., 2019). Although some mitigations have been observed through small-scale local upwelled or mixed cold water (Tkaachenko and Soong, 2017; Safaie et al., 2018; Davis et al., 2021), the overall situation remains concerning."*

2. Explanation of Buoy Data
The authors state that they have used SST data collected from three coastal buoys. They should provide locations of these buoys using geographical coordinates. I strongly recommend including a map to present the general location of these buoys.

*Done. Please see figure 1 and L111-114.*
*"Three sets of SST data, collected from three coastal buoys located at Chenggong, Linshan Cape, and Magong stations (Fig. 1), all maintained by Taiwan's Central Weather Administration (CWA), were employed to assess the effectiveness of linear*

*regressions and the STL. The Chenggong, Linshan Cape, and Magong stations are located on the eastern coast of Taiwan, the northern coast of Taiwan, and the coast of Penghu Island, respectively."*

[Figure]

*Figure 1: Topography and coastal line surrounding Taiwan. The blue dots denote three coastal buoys at Chenggong, Linshan Cape, and Magong stations, maintained by Taiwan's Central Weather Administration (CWA).*

3. Figures

For figures 1(a) and 2 authors should indicate the numerical values of slopes for each regression method. Preferably using the same colors of regression lines. This will help the readers to compare the regression estimates by examining figures without going back and forth through text and figures. For Raw SST, please use a different color from OLSR1.

*Revised as suggested. See Figures 2 and 3.*

[Figure]

*Figure 2: (a) Time series (2010-2023) of sea surface temperature from Chenggong coastal buoy stations and its long-term trend estimated using OLSR1, OLSR2, GMR, OR, and STL methods. (b) The seasonal and remainder components of the STL result. The vertical magenta lines and triangles denote the mean value of the time axis.*

[Figure]

*Figure 3: Time series (2010-2023) of sea surface temperature from (a) Linshan Cape Station and (b) Magong Station and their long-term trend estimated using OLSR1, OLSR2, GMR, OR, and STL methods. The vertical magenta lines and triangles*

*denote the mean value of the time axis.*

4. Methods

In Figure 1(b), the authors plot the remainder components. However, it is unclear in the methods section how they obtain these remainder components. Please explain this in detail in the methods sections.

*The conceptual description has been provided in L102-109. The methodology of STL involves numerous trivial and detailed operations, which are not suitably incorporated into the present manuscript. We've cited the paper of Cleveland et al. (1990) for readers interested in the details.*

5. A summary table.

The authors do a good job of comparing the outcomes of different regression methods within the text of the manuscript. Since the comparison of these regression methods is the core objective of this paper, the authors should highlight their findings appropriately. For this, I suggest using a table to summarize and compare the findings. This will provide readers the opportunity to go through the findings of this important work at one stop rather than scouring through the text for each regression method.

*Thanks. We have summarized the results of different methods in Table 1 of the manuscript.*

*Table 1: Summary of the $\widehat{b_1}$ (unit: °C/yr) estimated using general linear regression, STL, evenized SST, SST anomaly, and a combination of linear and sinusoidal fitting. The slope derived from linear fitting to the STL nonlinear curve (blue lines in Figures 2a, 3a, and 3b) represents the $\widehat{b_1}$ value of STL. As for the methods of evenized SST, SST anomaly, and combined linear and sinusoidal fitting, the representative $\widehat{b_1}$ is determined as the mean value during its stable period, marked by the black dashed lines in Figure 7 (6 months trimmed time).*

| | Methods of general linear regression | | | | STL | Method of evenized SST | Method of SST anomaly | Method of linear and sinusoidal fitting |
|---|---|---|---|---|---|---|---|---|
| | OLSR2 | GMR | OLSR1 | OR | | | | |
| Chenggong | 1.730 | 0.586 | 0.198 | 0.198 | 0.192 | 0.193 | 0.189 | 0.180 |
| Linshan Cape | 10.656 | 1.231 | 0.142 | 0.142 | 0.13 | 0.124 | 0.109 | 0.109 |
| Magong | 11.437 | 1.111 | 0.108 | 0.108 | 0.087 | 0.09 | 0.080 | 0.082 |

Minor Comments

1. L59-61: I assume μ is mean. But it will be helpful to readers if you define it properly.

*Corrected. See L79.*

2. L105: (2) should be (b)

*Corrected.*

3. L162: Recheck figure reference (Fig 3 or 4?)

*It is now Figure 5.*

4. L259-261: It seems the letter "O" is used instead of the number zero (0).

*We used degree (°).*

5. L299: change "real-sea" to "observed"

*Thanks. The associated sentence has been changed due to the other revision.*

---

## Author Response (AR2)

Responses to the Editor

I have received reviews from the two reviewers. One reviewer suggests including a legend (for plot lines) in Figure 7 (*See our response to the reviewer #2 below*). The other reviewer still has some concerns related to the comparison of new approach with the sinusoidal fitting approach, whose comments are attached below. If the new approach is not always better than the sinusoidal fitting approach, it is important to point out this limitation.

*Thank the Editor for the suggestions. We've replied to the reviewer #1 as follows. We also added some sentences to enhance the illustration about the performance of the fitting method and our method. See L389-394.*

*"While our method may have a more significant degree of uncertainty than linear and sinusoidal fitting, this uncertainty remains within an acceptable range. The fitting method generally performs better, as seen in Figure 8. However, linear and sinusoidal fittings can be unstable in occasional cases. The poor fitting may be addressed by providing better initial guess values, constraining parameter intervals, changing the numerical method, filtering the data, and other approaches. All of these require additional trials. Our proposed method provides another robust and efficient method that can avoid this disadvantage. Users can choose the method that best suits their analysis needs."*

Responses to Reviewer #1

I appreciate the authors' efforts in incorporating my previous suggestions and comparing their method with other widely accepted approaches in the atmospheric and climate community. Figures 7 and 8 in the revised manuscript are pivotal; however, I am somewhat surprised by the apparent uncertainty in the sinusoidal fitting. It would be beneficial if the authors could include a figure depicting a synthetic time series where the sinusoidal fitting fails and their proposed method excels.

*Thanks. The additional figure suggested by the review has been incorporated in the revised manuscript as Figure 9.*

*The associated sentences are in L335-342.*
*"Figure 9a shows a successful fitting curve of (5) (blue solid line), which overlaps with the simulated data (red solid line), when the data length used is 8 years. The resulting long-term trend (blue dashed line) also aligns with that from the evenized SST (red*

*dashed line, but it is exactly covered by the blue dashed line). However, unexpected fitting failures can cause large deviations (blue line in Fig. 8a), such as in the example when the data length of 7 years is used (Fig. 9b). The fitting curve (blue solid line in Fig. 9b) has a smaller seasonal amplitude and a clear phase shift compared to the simulated data (red solid line in Fig. 9b). The estimated slope of the long-term trend (blue dashed line) is gentler than the known trend. In contrast, the known trend agrees with that estimated using evenized SST (red dashed line)."*

[Figure]

*Figure 9: Linear and sinusoidal fitting curve of equation (5) using simulated data lengths of (a) 8 years and (b) 7 years. The red and blue solid lines represent the simulated data and its fitting curve, respectively. The red and blue dashed lines represent the long-term trends from the evenized SST and fitting methods, respectively.*

*We do not claim that our method is entirely superior to the combination of linear and sinusoidal fitting, which generally performs better, as seen in Figure 8. However, it is well known that small-scale variations and noise in the data can occasionally lead to poor fitting. Poor fitting may be addressed by providing better initial guess values, constraining parameter intervals, changing the numerical method, filtering the data, and other approaches. All of these require additional trials. We also added some associated sentences to strengthen our illustration in L335-342.*

Additionally, I suspect that the limitations in fitting accuracy may be due to the model only incorporating once-per-year cycles, whereas actual seasonal cycles might exhibit two or three cycle per year components. Previously, I had suggested experimenting with a larger number of cycles and employing an F-test to determine the necessary number of cycles. Demonstrating that their method maintains superior

stability compared to this enhanced sinusoidal fitting approach would strengthen their findings.

*Thanks for the further suggestions. The seasonal cycle typically shows a higher SST in summer and a lower SST in winter, resulting in a periodic variation of 365 days. Intraseasonal variations (two or three cycles per year) may be present in the data but usually have a much smaller amplitude than seasonal signals. Figure 8 suggests that a suitable data length for applying our method is longer than 7 years (i.e., 7 cycles). While it may not be as convincing as utilizing the F-test, it remains a good approach. Using synthetic and field data, we have examined various linear regression methods, identified a suitable approach to address the problem of seasonal bias, validated our method, and compared it with two traditional methods. There is always more to explore with the new method, particularly its applications to complex real-sea data other than around Taiwan. As the manuscript content has been a bit lengthy, we intend to save these further tests for our future efforts and possibly for other researchers in our community.*

Responses to Reviewer #2
Please consider including a legend (for plot lines) in Figure 7.

*Thanks. Adding a legend indicating what the lines and dots mean is a good idea. However, we had a hard time doing this because the lines in Figure 7 require longer explanations, which is different from the other figures. This makes the legend box occupy a considerable amount of space in the figure, making it too complex and busy. Therefore, we decided not to add the legend. We have illustrated the lines in the figure caption.*